



# How serious a problem is soil compaction in the Netherlands? A survey based on probability sampling.

Dick J. Brus[1] and Jan J.H. van den Akker[2]

[1]Biometris, Wageningen University, PO Box 16, 6700 AA Wageningen, the Netherlands
[2]Alterra, Wageningen University, PO Box 32, 6700 AA Wageningen, the Netherlands
*Correspondence to:* Dick J. Brus (dick.brus@wur.nl)

**Abstract.** Although soil compaction is widely recognized as a soil threat to soil resources, reliable estimates of the acreage of overcompacted soil and of the level of soil compaction parameters are not available. In the Netherlands data on soil compaction were collected at 128 locations selected by stratified random sampling. A map showing the risk of soil compaction in five classes was used for stratification. Measurements of bulk density, porosity, clay content and organic matter content were used to compute the relative bulk density and relative porosity, both expressed as a fraction of a threshold value. A soil was classified as overcompacted if either the relative bulk density exceeds 1 or the relative porosity is below 1. The sample data were used to estimate the means of the two soil compaction parameters and the areal fraction overcompacted. The estimated global means of relative bulk density and relative porosity were 0.946 and 1.090, respectively. The estimated areal fraction of the Netherlands with overcompacted soils was 45 %. The estimates per risk map unit showed two groups of map units, a 'low risk ' group (unit 1 and 2, covering only 4.6 % of the total area) and a 'high risk' group (unit 3, 4 and 5). The estimated areal fraction overcompacted soil was 0 % in the 'low risk' unit and 47 % in the 'high risk' unit. The map contains no information about where overcompacted soils occur. This was caused by the poor association of the risk map units 3, 4 and 5 with the soil compaction parameters and soil overcompaction. This can be explained by the lack of time for recuperation.

## 1 Introduction

Soil compaction is recognized as one of the major soil threats. Van Camp et al. (2004) recognized soil compaction as one of the eight soil threats requiring further attention. In 2006 the Thematic Strategy for Soil Protection was launched by the European Commission (European Commission, 2006). Subsoil compaction is of more concern than topsoil compaction, because of its persistency (Alakukku, 2000; Berisso et al., 2012, 2013). Subsoil compaction is defined as compaction of the soil below the cultivated layer. This compacted layer is referred to as the panlayer, hardpan or plow pan. The panlayer is often the bottleneck for the functioning of the subsoil, because it is denser and less permeable for roots, water and oxygen than the soil below this layer.

Lipiec and Hatano (2003) review indices and methods to quantify the effects of compaction on soil physical properties and crop growth. They concluded among others that yield decrease in overcompacted soil is frequently attributed to excessive mechanical impedance, reduced water infiltration and crop water use efficiency, insufficient aeration or their combination





depending on weather conditions. Etana et al. (2013) stressed the impact of subsoil compaction on preferential flow of water in a sandy clay soil, which can result in a fast transport of nutrients and agrochemicals to deeper soil layers and ground water. Schjonning et al. (2015) present an overview of results of field experiments on crop yield reduction by subsoil compaction. Alblas et al. (1994) report average yield reductions of silage maize on sandy soils with a compacted subsoil of 15 % with

a wheel load of 5 Mg and 4 % with a wheel load of 2.5 Mg. Hakansson and Reeder (1994) report 2.5 % permanent yield reductions in long term experiments with wheel loads of 5 Mg applied in the first year of the experiment. After this first year wheel loads were limited to 2 Mg to prevent further compaction. The same kind of long term experiments by Voorhees (2000), however, with wheel loads of 9 Mg resulted in permanent yield reductions of on average 6 %. It should be noted that in practice wheel loads of 5 to 9 Mg or even higher are commonly used in heavy mechanized agriculture during manuring and harvesting.

Hakansson and Reeder (1994) also studied the recuperation of soil compaction in a clay loam soil. In the first 5 years the topsoil recuperated to a great extent. In the first 10 years also the upper part of the subsoil to a depth of about 40 cm recuperated considerably, however in the third layer below 40 cm depth the recuperation was almost zero and caused a permanent yield reduction of 2.5 %.

Soil compaction is estimated to be responsible for the degradation of an area of about 33 million ha in Europe (Van Ouwerk-

erk and Soane, 1994). About 32 % of the subsoils in Europe is highly vulnerable and another 18 % is moderately vulnerable to subsoil compaction (Fraters, 1996). However, these are very rough estimates and not the result of a thorough assessment. Jones et al. (2003) present a map of the vulnerability of subsoils to compaction. The authors concluded that at the moment on the basis of the existing information, any attempt to identify the vulnerability to compaction of subsoils in Europe, on a spatial basis, lends itself to fundamental improvement. Also the assessment of the compaction state of subsoils is scarce and

incomplete and requires improvement (van den Akker et al., 2003).

Previous work by van den Akker and Hoogland (2011) was not conclusive about how serious the problem is in the Netherlands either. Two risk-assessment methods were used to map the vulnerability and susceptibility to soil compaction. These maps were compared to a map showing the probability that the subsoil is already compacted. The agreement of the vulnerability and susceptibility maps with the probability map was poor. The probability of compacted soil was mapped using legacy

data on bulk density in the Dutch Soil Information System. The value of these data for assessing the current soil compaction is restricted because most of the measurements were done more than 20 years ago. Another problem was that the sampling locations were not selected by probability sampling. For that reason the only option was to construct the map and estimate the areal fraction overcompacted soil by a model-based approach, more specific space–time kriging. The available data for the calibration of the model were rather scarce, so that the quality of the geostatistical model is questionable. This together with

the questionable quality of the legacy data was the motivation for a new, nationwide survey, specifically designed to quantify how serious the problem of current soil compaction is in the Netherlands.

The aim of this research was to design a sample for estimating the current means of soil compaction parameters and the areal fraction where soil compaction has exceeded a critical threshold. These means and areal fraction must be estimated for the Netherlands in its entirety, as well as for the five units of the soil compaction risk map. The estimates must be accompanied

with estimates of their accuracies.





The soil compaction risk map for the Netherlands of van den Akker et al. (2013) plays a central role in this study. Therefore we first describe how van den Akker et al. (2013) constructed this map. In the subsequent section we describe the methodology of this study.

## 2 Soil compaction risk classification and mapping

5 The risk of soil compaction is a function of wheel loads of machines which is related to landuse, and soil mechanical strength which is determined by various soil properties such as soil texture and water content. The map of soil compaction risk was constructed by combining information derived from the landuse database of the Netherlands (Hazeu et al., 2010), and from the Soil Map of the Netherlands 1:50,000 and associated database with descriptions of typical soil profiles (de Vries, 1999).

The landuse database was used to determine typical agricultural machinery and associated typical wheel loads, tyres and 10 tyre inflation pressures for agricultural areas in The Netherlands. In this an inventory of Vermeulen et al. (2013) was used, in which typical, commonly used heavy machinery, wheel loads and tyres in 1980 and 2010 are compared. The SOCOMO model (van den Akker, 2004) was used to calculate for each of these wheel loads the soil stresses at several depths.

The calculated soil stresses for 2010 were compared with the soil strengths in the same way as presented in van den Akker (2004) and van den Akker and Hoogland (2011). Also the same soil classification as in van den Akker (2004) and van den 15 Akker and Hoogland (2011) was used. Based on the landuse map and the soil map 1 : 50 000 of The Netherlands for each parcel the exerted soil stresses on the subsoil by typical wheel loads for that land use were compared with the strength of that subsoil for a wet soil (at about field capacity) and a moist soil (a soil water suction of about -30 kPa). Five risk categories were considered: very high, high, moderate, low and very low. If the exerted soil stresses were higher than the strength of a moist soil, then the risk of subsoil compaction was considered to be "high". If the exerted soil stresses did not exceed the strength 20 of the moist soil, however, exceeded the strength of the wet soil, then the subsoil compaction risk was "moderate". In case the exerted soil stresses didn't exceed the strength of the wet subsoil, then the subsoil compaction risk was "very low".

In a second step factors that increase or decrease the risk of subsoil compaction on the long term were taken into account. Factors that improve the resilience and natural recuperation of the compacted subsoil and in that way decrease the subsoil compaction risk are:

25    – The soil is well drained and in general dry, improving the resilience and the natural recuperation

   – Clay content > 17.5 %: improved natural recuperation by swelling and shrinkage and structure forming processes

   – Organic matter content > 4 %: improved rebound after loading, biological structure forming processes

   – Coarse sand: hardly any increase in density, water infiltration is never a problem

   – Only a limited part of the parcel can be trafficked, so compacted: e.g. forests or orchards

30 Factors that increase the risk of subsoil compaction are:



- The soil is often wet

- The typical wheel loads of the land use will cause compaction at depths $> 40$ cm.

All positive and negative factors are added together and the risk class in the first step is increased or decreased with a maximum of one class. The change in class is limited to one step to account for the fact that overloading and compaction of the subsoil is cumulative in time and recuperation by shrinkage and biological processes is never complete, therefore the risk classification should be mainly determined as a function of the exerted stresses at a certain depth and the strength of the soil at that depth. Figure 1 shows the soil compaction risk map.

## 3 Sampling theory

### 3.1 Sampling design

For estimating means of soil compaction parameters, locations were selected by probability sampling, i.e. by random sampling with known inclusion probabilities which are $> 0$ for all locations in the study area (Särndal et al., 1992). With probability sampling model-free, design-based estimates of spatial means and their variances can be obtained, so that discussions on the validity of the results are avoided (de Gruijter and ter Braak, 1990; Brus and de Gruijter, 1997). Stratified simple random sampling was chosen as a design-type (de Gruijter et al., 2006). For stratification we used the map showing the risk of soil compaction in five classes (Figure 1). When the risk map units are related to the soil compaction parameters measured in this study (see hereafter), we expect a gain in precision of the estimated nationwide means compared to simple random sampling with the same sample size. Besides, a map showing the provinces of Zeeland, Noord-Brabant, Gelderland and remaining provinces was used for stratification This map was used to control the sample sizes in these administrative units. The three mentioned provinces contributed additional financial resources, so that these provinces claimed extra sampling locations. The assumption was that in the provinces of Gelderland, Noord-Brabant and Zeeland the problem of soil compaction is more serious, due to the intensive use of heavy machines in agriculture. The ultimate strata were obtained by overlaying the two maps. All five risk classes were present in all administrative units, so that the total number of strata became $5 \times 4 = 20$.

The total sample size was 128. The sample sizes in the provinces Gelderland, Noord-Brabant, Zeeland were 20, 39, 30, respectively, leaving 39 for the remaining provinces. These sample sizes were allocated proportionally to the area of the five risk map units within the provinces. The total sample sizes in the risk map units 1 (low risk) to 5 (high risk) were 4, 5, 56, 44 and 19, respectively. The small sample sizes for the risk map units 1 and 2 reflect the small areas of these two units: the sum of their areas is only 4.6 % of the total area.

The target population consists of all soils in the Netherlands, both cultivated and uncultivated soils, except soils with a low compaction risk due to peat layers, naturally compacted soils ('knipkleigronden') and soils in glasshouses.





## 3.2 Field sampling and laboratory measurements

The randomly selected locations were localized by differential GPS. If a randomly selected sampling location was unsuitable for collecting soil samples (no soil present, no permission, not part of the target population), the first point on a reserve list, in the same stratum as the omitted point, was added to the list of points to be visited.

At each sampling location three volumetric soil samples were collected using a cylinder with a diameter of 7.6 cm. The length of the soil cores was 5 cm. The soil cores were collected directly below the plough layer (sandy soils below 35 cm, clay soils below 20 to 22 cm). The clay content and soil organic matter content was estimated by the soil surveyor in the field. The dry bulk density and the actual moisture content was determined in the laboratory by weighing and drying of the samples. The porosity was calculated from the dry bulk density using a specific weight of the mineral parts of 2.65 g.cm$^{-3}$ and a specific

weight of the soil organic matter of 1.47 g.cm$^{-3}$.

## 3.3 Soil compaction parameters

We used as soil compaction parameters the relative bulk density and relative porosity. The relative bulk density is defined as the actual bulk density as a fraction of the threshold value of the bulk density (van den Akker and Hoogland, 2011). For sand and loamy soils (clay content $< 16.7$ %) this threshold value is 1.6 g cm$^{-3}$; for soils with clay content $> 16.7$ % the threshold

value is $1.75 - 0.009 \times clay$ g cm$^{-3}$.

    The relative porosity is defined as the actual porosity as a fraction of the threshold value of the porosity, which is 0.4 as determined in the ENVASSO project (Huber et al., 2008). In general this threshold value was only a problem in sandy and loamy soils with some organic matter.

    If either the relative bulk density $> 1$ or the relative porosity $< 1$, the soil is classified as being overcompacted.

## 20 3.4 Estimation of means and areal fractions

The global means of the relative bulk density and relative porosity were estimated by design-based inference, more specifically by the usual estimator for stratified simple random sampling:

$$\hat{\bar{y}} = \sum_{h=1}^{H} w_h \hat{\bar{y}}_h$$

$$\hat{\bar{y}}_h = \frac{1}{n_h} \sum_{i=1}^{n_h} y_{hi} \tag{1}$$

with $H$ total number of strata ($H = 20$), $w_h$ the weight of stratum $h$ quantified by the relative area, $\hat{\bar{y}}_h$ the estimated mean of stratum $h$, $n_h$ the number of sampling points in stratum $h$, and $y_{hi}$ the measurement of the target soil property at location $i$ in stratum $h$. The areal fraction overcompacted can be estimated by the same equations, replacing $y_{hi}$ by an indicator having value 1 if the soil at that location is overcompacted and 0 else.





These estimators were also used to estimate the means of the two soil compaction parameters and the areal fraction over-compacted for the five units of the soil compaction risk map and for the three provinces. These subareas are unions of complete strata, i.e. they do not contain one or more strata which only partly belong to the subarea, so that estimation is straightforward.

### 3.4.1 Estimation of sampling variances

In all four strata of risk map unit 1 and three strata of risk map unit 2 only 1 point was selected. This complicates the estimation of the sampling variance of the estimated means. Following the approach of Cochran (1977), we collapsed all four strata of risk map unit 1, and all four strata of risk map unit 2. The total number of sampling points in the two collapsed strata were four (risk map unit 1) and five (risk map unit 2). After collapsing the total number of strata was $3 \times 4 + 2 = 14$. The sampling variance of the estimated means was then estimated by

$$
\widehat{V}(\hat{\bar{y}}) = \sum_{c=1}^{C} w_c^2 \widehat{V}(\hat{\bar{y}}_c)
$$

$$
\widehat{V}(\hat{\bar{y}}_c) = \frac{s_c^2}{n_c}
$$

$$
s_c^2 = \frac{1}{n_c - 1} \sum_{i=1}^{n_c} (y_{ci} - \hat{\bar{y}}_c)^2 \tag{2}
$$

with $C$ total number of strata after collapsing ($C = 14$), $w_c$ the weight of (collapsed) stratum $c$ quantified by the relative area, $\widehat{V}(\hat{\bar{y}}_c)$ the estimated sampling variance of the estimated mean of (collapsed) stratum $c$, $s_c^2$ the estimated spatial variance within (collapsed) stratum $c$, $n_c$ the number of sampling points in (collapsed) stratum $c$, and $\hat{\bar{y}}_c$ the estimated mean in (collapsed) stratum $c$ (the sample average in (collapsed) stratum $c$). Standard errors of the estimated means are computed by the square root of the estimated sampling variances.

The sampling variance of the estimated areal fractions was estimated by (Cochran, 1977)

$$
\widehat{V}(\hat{\bar{y}}) = \sum_{c=1}^{C} w_c^2 \frac{\widehat{V}(\hat{\bar{y}}_c)}{n_c - 1}
$$

$$
\widehat{V}(\hat{\bar{y}}_c) = \hat{\bar{y}}_c (1 - \hat{\bar{y}}_c) \tag{3}
$$

## 4 Results and Discussion

### 4.1 Descriptive statistics of data

Figure 2 shows boxplots of the relative bulk density and relative porosity for the five risk map units. The lower and upper side of the box represent the first and third quartile, the central line the median. Note that the boxplots for risk map unit 1 and 2 are based on 4 and 5 measurements only. For both soil compaction parameters the risk map units can be aggregated into two distinct groups, a group with relatively low soil compaction consisting of map units 1 and 2, and a group of relatively high soil compaction consisting of map units 3, 4 and 5. Differences between risk map units within the same group were small




compared to differences between groups. In all three risk map units 3, 4 and 5 outliers occurred with a relatively small relative bulk density and relatively large relative porosity (dots in Figure 2).

## 4.2 Means of soil compaction parameters and areal fraction overcompacted

The estimated global mean of relative bulk density was 0.946 with an estimated standard error of 0.012. The estimated global
mean of relative porosity was 1.090 with an estimated standard error of 0.020. The estimated areal fraction of the Netherlands with overcompacted soils was 0.446 with an estimated standard error of 0.053.

Design-based estimates of the means of the two soil compaction parameters and of the areal fractions overcompacted per risk map unit are shown in Figure 3. The error bars represent the standard error of the mean. So for a 95 % confidence interval the length of the bars must approximately be doubled. The estimated means confirm what we have seen in the raw boxplots
(Figure 2). Estimated means of relative bulk density were relatively low in map units 1 and 2 and relatively high in map units 3 to 5, and accordingly estimated means of relative porosity were relatively large in map units 1 and 2 and relatively small in maps units 3 to 5. The standard errors of the estimated means for map units 3 to 5 were acceptable; for map units 1 and 2 these were large compared to the estimated means due to the very small sample sizes. The error bars of map units within above mentioned groups clearly overlap, so that without statistical testing we can safely conclude that the means of risk map units
within a group were not significantly different.

For map units 1 and 2 the estimated areal fractions overcompacted soils were both 0 (in both units no sampling points had a relative bulk density $> 1$ or a relative porosity $< 1$), whereas for map units 3 to 5 these varied from 0.34 (map unit 4) to 0.56 (map unit 3).

As differences between map units 1 and 2, and between the map units 3, 4 and 5 were small, we also estimated means and
areal fractions for these two groups (Table 1). The sample sizes in these two groups were 9 (map units 1 and 2) and 119 (maps units 3, 4 and 5). The estimated mean relative bulk density in the 'high risk' group (units 3, 4 and 5) was 9.2 % larger than in the 'low risk' group. The difference in estimated mean relative porosity between the two groups was larger: 1.07 for the 'high risk' group of map units versus 1.42 for the 'low risk' group. Note that the mean relative porosity for the 'high risk' group exceeded value 1. The areal fraction overcompacted was about 47 % for the 'high risk' group, whereas it was 0 for the 'low
risk' group. All differences were significant at a significance level of 0.01.

Finally, we estimated means of the two soil compaction parameters and areal fraction overcompacted for the 'high risk' group of map units inside the three provinces, to check the assumption that in these provinces the problem of soil compaction was more serious (Figure 4). The means of relative bulk density and relative porosity indicated more serious soil compaction problems in these provinces indeed, although the differences with the global means were not significant. The estimated areal
fraction overcompacted was larger than the global areal fraction for the provinces of Noord-Brabant and Gelderland, but not for Zeeland.

The aggregated map unit 'high risk' covers 95.4 % of the Netherlands. About 47 % of the subsoils within this aggregated map unit are overcompacted, but the map contains no information about where these overcompacted subsoils occur, as the risk



map units 3, 4 and 5 are not associated with the subsoil compaction parameters and subsoil overcompaction, i.e. differences between the map units 3, 4 and 5 of the current means of soil compaction parameters were small.

A possible explanation is the poor quality of the soil compaction risk map. The soil compaction risk class as depicted on the map will not correspond everywhere with the risk class in the field, i.e. the risk class as based on the soil profile characteristics

observed in the field. We estimated the purity of the five map units, i.e. the areal fractions of the map units where the soil compaction risk class as depicted on the map corresponds with the risk class in the field (Brus et al., 2011). For map units 1 and 2 the estimated purity was 1, but these estimates were based on a few sampling points only, and therefore are very inaccurate. For map units 3, 4 and 5 the estimated purities were 0.80, 0.71 and 0.84, respectively. This indicates that the small differences in soil compaction parameters between the map units cannot be attributed to low map unit purities. This was confirmed by the

estimated means of the soil compaction parameters for the risk classes in the field 5. The patterns are very similar to those for the risk map units (3. Again the differences between the risk classes 3, 4 and 5 in the field were small.

A second explanation could be a poor performance of the SOCOMO model. However, comparisons between modeled and measured stresses showed good agreement (van den Akker, 2004; Keller et al., 2014). It should also be noted that in general the calculated stresses were much higher than the strength of the subsoil (van den Akker et al., 2013), and also much higher

than the strength threshold value of 40 kPa for the subsoil determined by Keller et al. (2012).

A third possible explanation is the lack of time for natural recuperation of subsoil compaction. Due to the intensive agricultural land use the subsoil is overloaded every second or third year, so considering a recuperation time of about 10 years of the upper subsoil up to a depth of 40 cm (Hakansson and Reeder, 1994), the expected natural recuperation in clay subsoils or sandy subsoils with a soil organic matter content $> 4$ % can only be very limited and temporally.

The 47 % of the area with 'high risk' of subsoil compaction that has indeed an overcompacted subsoil is in good agreement with the 50 % overcompacted subsoils predicted for 2010 in van den Akker and Hoogland (2011). This prediction was based on legacy data mainly collected before 1988, whereas the data of this paper were collected by probability sampling in 2013.

## 5   Conclusions

– About 45% of the soils in the Netherlands are overcompacted.

– The map of risk for subsoil compaction of van den Akker et al. (2013) provides only very rough information about where these overcompacted subsoils occur in the Netherlands.

– In terms of the soil compaction parameters relative density and relative porosity, and in terms of the areal fraction overcompacted soil only two risk classes and risk map units can be distinguished: 'low risk' (risk classes/map units 1 and 2) and 'high risk' (risk classes/map units 3, 4 and 5).

– Lack of time for natural recuperation can be an explanation for the fact that, despite the good quality of the risk map in terms of map unit purity and class representation, no differences in subsoil compaction can be distinguished between the map units 3, 4 and 5.



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



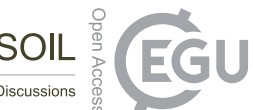

**Table 1.** Design-based estimates of means of two soil compaction parameters and of areal fraction overcompacted, for the groups 'low risk' (map units 1 and 2) and 'high risk' (map units 3, 4 and 5). Between brackets: standard error

|  | low risk | high risk | p-value |
| --- | --- | --- | --- |
| relative bulk density | 0.858 (0.0276) | 0.950 (0.0125) | 1.3E-3 |
| relative porosity | 1.42 (0.0666) | 1.07 (0.0206) | 2.9E-7 |
| areal fraction overcompacted | 0.00 (0.000) | 0.467 (0.0551) | 1.1E-17 |



**Figure 1.** Map of risk for soil compaction (van den Akker et al., 2013)

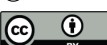


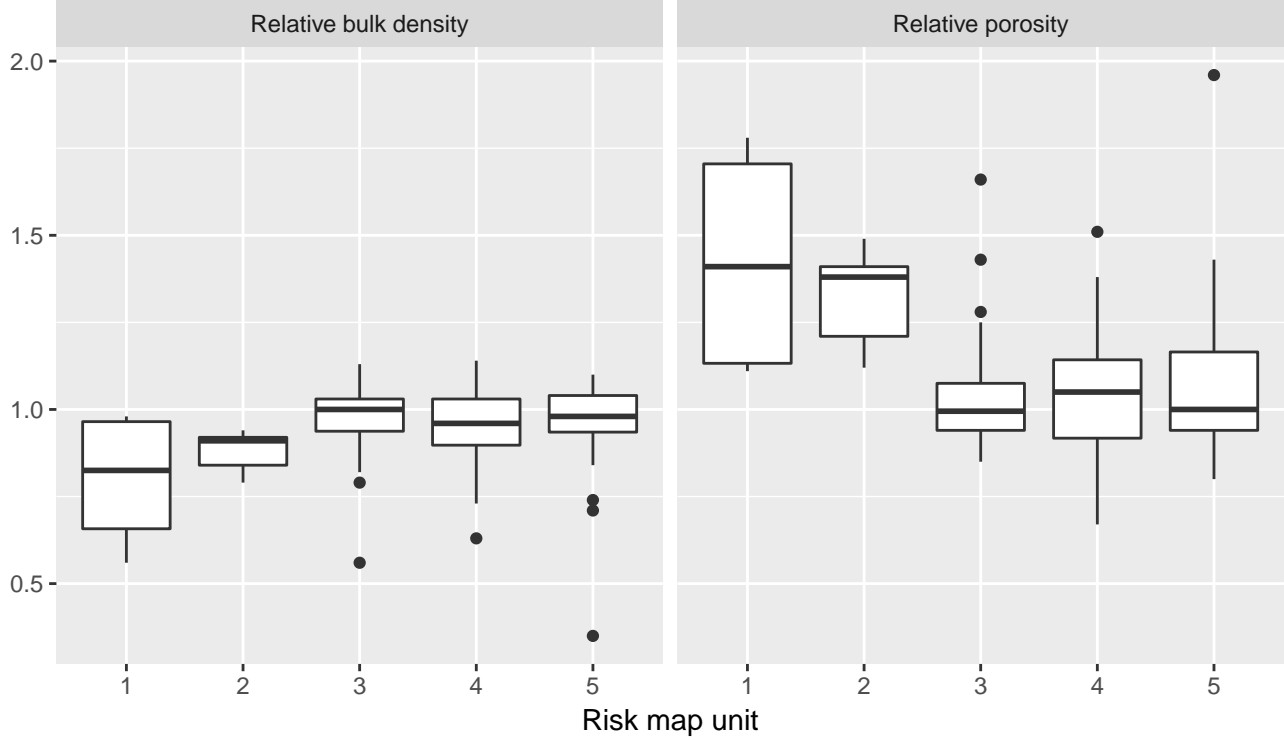

**Figure 2.** Boxplots of relative bulk density and relative porosity per risk map unit (1: low risk; 5: high risk)



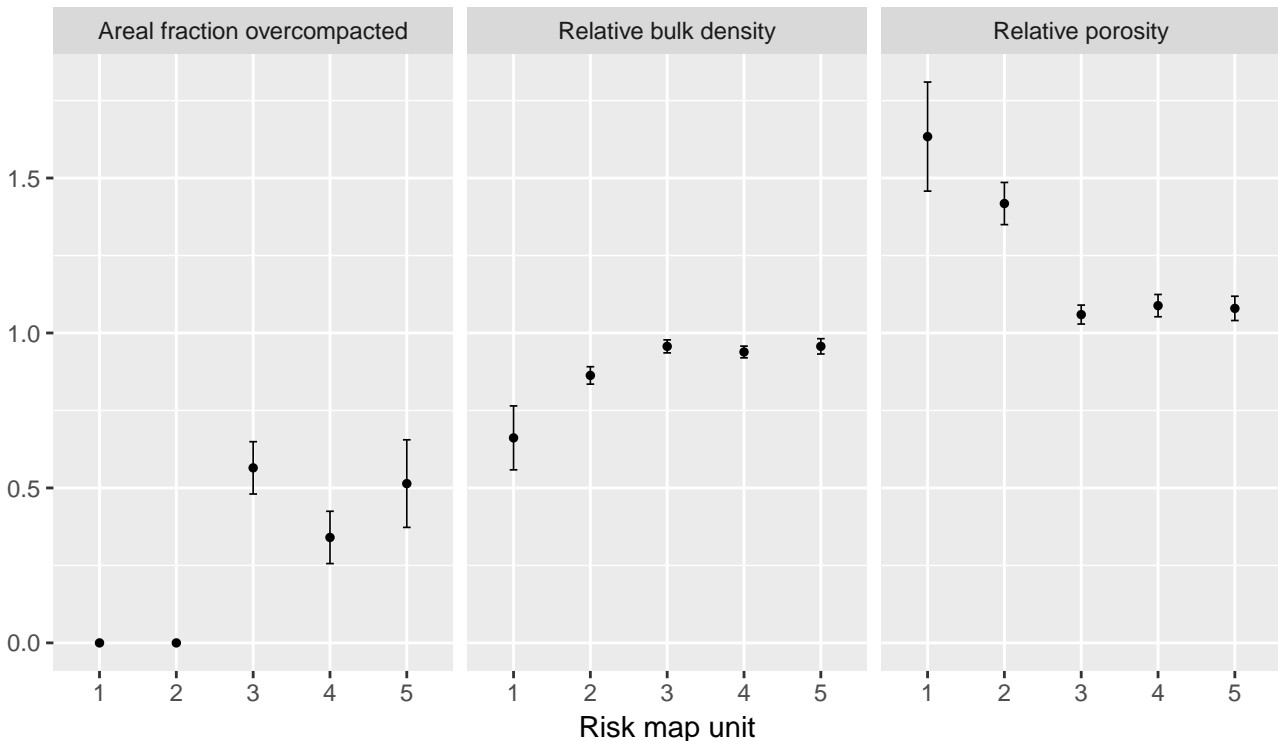

**Figure 3.** Estimated means of soil compaction parameters and areal fractions overcompacted for the five risk map units. The error bars indicate the standard error of the estimated means or areal fraction




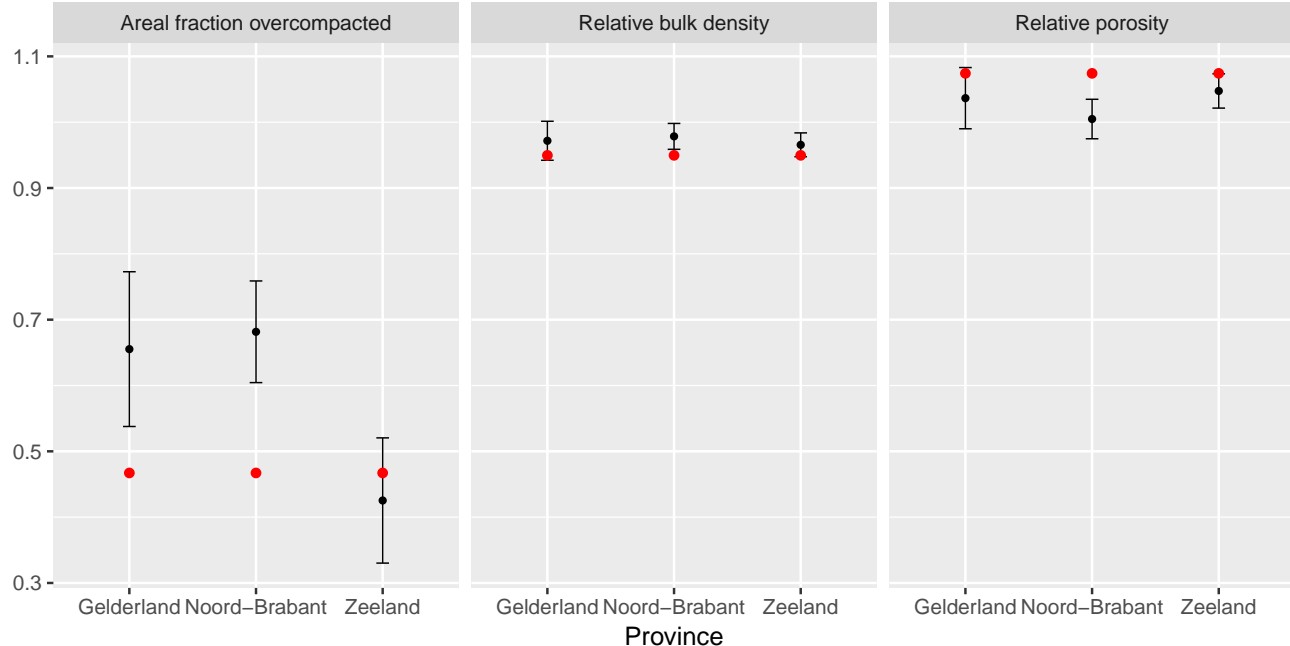

**Figure 4.** Estimated means of soil compaction parameters and areal fractions overcompacted for 'high risk' group of map units (units 3, 4 and 5) inside the three provinces. The error bars indicate the standard error of the estimated means or areal fraction. The red dots are the estimated means or areal fraction for the Netherlands.



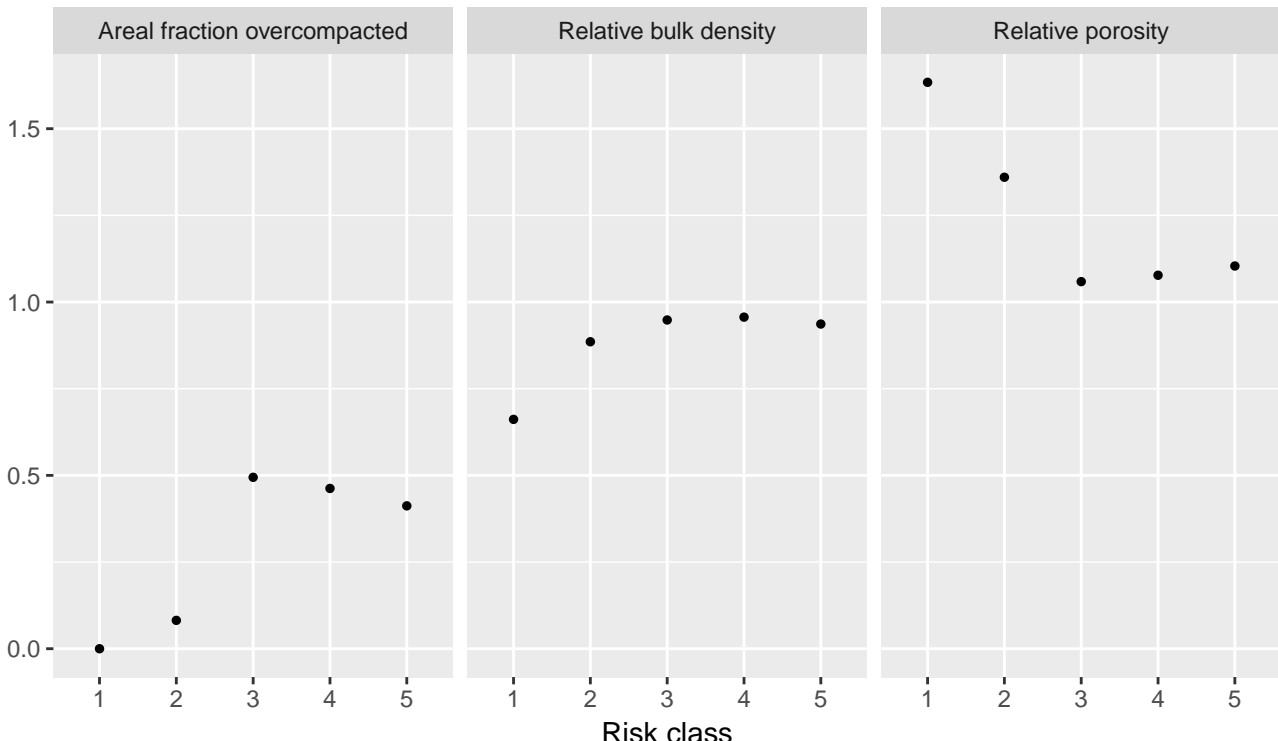

**Figure 5.** Estimated means of relative bulk density and relative porosity and estimated areal fraction overcompacted for the risk classes in the field.