# Peer review of "How serious a problem is subsoil compaction in the Netherlands? A survey based on probability sampling."

_SOIL, 2017_

## Referee Comment (RC1) · Anonymous Referee #1 · 4 Nov 2017

General comment This manuscript presents interesting results showing soil compaction risk classes and risk map units based on relative porosity and relative bulk density in Dutch soils. It is relevant to SOIL and of interest to a wider audience. The manuscript require amendments and explanations that have been indicated below:

1. As soil samples were taken directly below the plough layer (subsoil) (P 5 L 5-6) suggest changing soil to subsoil. Also "subsoil" is frequently used in the text. 2. Provide information about whether soil (subsoil) compaction affects crop yields and other soil function in the studied areas. 3. P4, L 9-10 and some other places: change specific density to particle density. 4. P 7 L 26 and some other places including captions to

Figures: "soil" is used here. Does it mean "subsoil" from which soil cores for determination of bulk density and porosity were taken as indicated in P 5 L 5-6). It should be clarified.

Minor points P 2, L 3 and elsewhere: Schjonnong > Schjønnong. P 9, L 30 and elsewhere: Hakansson > Håkansson.

---

## Short Comment (SC1) · 1 Dec 2017

Dear authors,

the title of your manuscript grabbed my attention. Studying the areal extent of compacted agricultural land is of great importance. However, it is a challenging task and various approaches have been suggested to do so.

While reading the present manuscript, the following issues came into my mind:

The ratio of bulk density (porosity) and respective threshold values served as indicators to classify subsoil as compacted or not compacted. The reported threshold values were

underpinned with citations. However, when briefly screening the respective articles, I could not find information on how the threshold values were exactly derived and related to soil functions. I would therefore recommend to explain the origin of these threshold values in a revised version of this manuscript in greater detail.

The study concludes that about 45% of the soils in the Netherlands are overcompacted. However, this confused me a little because in the methods section it says that organic soils and naturally compacted soils were omitted from the present study (page 4, line 28). What's the areal percentage of organic and naturally compacted soils in the Netherlands? Furthermore, mentioning the existence of naturally compacted soils is an important point, which I believe would be interesting to be addressed in greater detail in a revised version of this manuscript. For example Gao et al. (2016) pointed out that bulk density (or the degree of soil compaction) typically increases with soil depth merely due to the overburden pressure exerted by the above soil column. Could the present results be used to distinguish between anthropogenic and natural soil compaction?

Finally, I was wondering at which depth soil samples were taken if no plough layer was present (if I understood correctly also uncultivated soils were sampled, right? page 4, line 28) and whether on cropland typically more compacted parts of the fields like traffic lanes or headland were sampled as well.

Best regards,

Florian Schneider

—-

Gao, W., et al. "Deep roots and soil structure." Plant, cell & environment 39.8 (2016): 1662-1668.

---

## Referee Comment (RC2) · Anonymous Referee #2 · 16 Dec 2017

Dear Authors I appreciate the mammoth amount of effort a nationwide soil survey will have taken and the importance for such surveys. The data obtained is important and could be compared to surveys from other Nations. It is wise to get a baseline appreciation on the widespread nature of compaction, which perhaps is a chronic problem for yields in European soils. I agree with the other comments already highlighted by the other reviewers and suggest they are taken on board. However, I think the manuscript is of publishable quality and of a wide interest to the readership of this journal. Here are a few amendments P1L20: Do you mean the layer above. P3 L28: do you mean particle density? P8 L27: I think when you refer to just density here and elsewhere you are referring to bulk density, this may go against the comment by reviewer 1.

---

## Author Comment (AC1) · 20 Dec 2017

First we would like to thanks reviewer 1 for her/his constructive comments.

1. We agree with this point, and changed soil into subsoil at all places where it refers to our study

2. We did not register the crop yields at the selected sampling sites. Note that to study the effect of subsoil compaction on crop yields an experimental design is needed. For an observational study into the effect of subsoil compaction on crop yields a different sampling design is needed: selection of pairs of sites, with/without subsoil compaction,

comparable with respect to other factors that control crop yield (such as nutrient and water availability). This was not the aim of this research

3. Thanks, we replaced specific weight by particle density

4. See comment 1, so we replaced soil by subsoil where appropriate

Minor points: we corrected the spelling of the Scandinavian authors (sorry, I did not know how to do this in LaTeX, now I know)

---

## Author Comment (AC2) · 20 Dec 2017

Thanks for the kind words about our manuscript, and for the valuable comments.

P1, L 20: the word below is correct. We are talking about the subsoil here (see first part of the sentence) , and the panlayer is at the top of the subsoil. The panlayer is less permeable than the soil below it.

P3, L28: density refers here to dry bulk density, so we replaced density by dry bulk density

P8, L27: correct, relative density is relative dry bulk density.

---

## Author Comment (AC3) · 21 Dec 2017

Dear Florian, many thanks for your valuable remarks and questions. Below some answers.

We will give some more detail about the threshold in the revised paper

Then your question about the 45% overcompacted soils. This is 45% of the target population, and so it will be slightly smaller for the entire Netherlands, see Figure 1 (grey area) which part of the Netherlands was excluded. The target population excludes:

a. build-up areas, glasshouses, infrastructure (roads), water

b. soils with peat at about 20 to 25 cm below surface. So peat soils with a mineral topsoil are NOT excluded.

c. naturally compact subsoils, such as boulder clay soils, 'knipkleigronden' and others

The area of the soils of category b is about 4.5%. The estimated total area (a, b and c) is about 5% to 6% of the area of the Netherlands.

Then your question about the sampling depth when no plough layer was present, for instance in uncultivated soils. When it was not clear at what depth the subsoil was compacted, both in uncultivated and cultivated soils the decision was based on the measurements with the penetration resistance as measured with a penetrometer.

Finally your question about traffic lanes and headland of arable fields. Sampling sites were randomly selected, and selected sites in these parts of the fields were NOT excluded.

I hope this answers your questions.

best wishes

Dick Brus

---

## Author Response (AR1)

**Anonymous Comment Referee 1**

General comment This manuscript presents interesting results showing soil compaction risk classes and risk map units based on relative porosity and relative bulk density in Dutch soils. It is relevant to SOIL and of interest to a wider audience. The manuscript require amendments and explanations that have been indicated below:

1. As soil samples were taken directly below the plough layer (subsoil) (P 5 L 5-6) suggest changing soil to subsoil. Also "subsoil" is frequently used in the text.

We agree with this point, and changed soil into subsoil at all places where it refers to our study.

2. Provide information about whether soil (subsoil) compaction affects crop yields and other soil function in the studied areas.

We did not register the crop yields at the selected sampling sites. Note that to study the effect of subsoil compaction on crop yields an experimental design is needed. For an observational study into the effect of subsoil compaction on crop yields a different sampling design is needed: selection of pairs of sites, with/without subsoil compaction, comparable with respect to other factors that control crop yield (such as nutrient and water availability). This was not the aim of this research.

3. P4, L 9-10 and some other places: change specific density to particle density.

Thanks, we replaced specific weight by particle density.

4. P 7 L 26 and some other places including captions to Figures: "soil" is used here. Does it mean "subsoil" from which soil cores for determination of bulk density and porosity were taken as indicated in P 5 L 5-6). It should be clarified.

See comment 1, so we replaced soil by subsoil where appropriate.

Minor points P 2, L 3 and elsewhere: Schjonnong > Schjønnong. P 9, L 30 and elsewhere: Hakansson > Håkansson.

We corrected the spelling of the Scandinavian authors (sorry, I did not know how to do this in LaTeX, now I know).

**Anonymous Comment Referee 2**

Dear Authors I appreciate the mammoth amount of effort a nationwide soil survey will have taken and the importance for such surveys. The data obtained is important and could be compared to surveys from other Nations. It is wise to get a baseline appreciation on the widespread nature of compaction, which perhaps is a chronic problem for yields in European soils. I agree with the other comments already highlighted by the other reviewers and suggest they are taken on board. However, I think the manuscript is of publishable quality and of a wide interest to the readership of this journal. Here are a few amendments

P1 L20: Do you mean the layer above.

The word below is correct. We are talking about the subsoil here (see first part of the sentence) , and the panlayer is at the top of the subsoil. The panlayer is less permeable than the soil below it.

P3 L28: do you mean particle density?

Density refers here to dry bulk density, so we replaced density by dry bulk density.

P8 L27: I think when you refer to just density here and elsewhere you are referring to bulk density, this may go against the comment by reviewer 1.

This is correct, relative density is relative dry bulk density.

**Short comment by Florian Schneider**

Dear authors,

the title of your manuscript grabbed my attention. Studying the areal extent of compacted agricultural land is of great importance. However, it is a challenging task and various approaches have been suggested to do so. While reading the present manuscript, the following issues came into my mind:

The ratio of bulk density (porosity) and respective threshold values served as indicators to classify subsoil as compacted or not compacted. The reported threshold values were underpinned with citations. However, when briefly screening the respective articles, I could not find information on how the threshold values were exactly derived and related to soil functions. I would therefore recommend to explain the origin of these threshold values in a revised version of this manuscript in greater detail.

The paper of van den Akker and Hoogland (2011), which is cited, gives details about the derivation of the threshold values; maube a problem was that the hyperlink in this paper of van den akker and Hoogland was not working anymore. For that reason we have added an updated hyperlink, as well as a reference to the reports I and V of the ENVASSO project, see p.5, lines 16 and 17.

The study concludes that about 45% of the soils in the Netherlands are overcompacted. However, this confused me a little because in the methods section it says that organic soils and naturally compacted soils were omitted from the present study (page 4, line 28). What's the areal percentage of organic and naturally compacted soils in the Netherlands? Furthermore, mentioning the existence of naturally compacted soils is an important point, which I believe would be interesting to be addressed in greater detail in a revised version of this manuscript. For example Gao et al. (2016) pointed out that bulk density (or the degree of soil compaction) typically increases with soil depth merely due to the overburden pressure exerted by the above soil column. Could the present results be used to distinguish between anthropogenic and natural soil compaction?

Good question! We understand the confusion. The 45% overcompacted soils refers to the target population, and so it will be slightly smaller for the entire Netherlands, see Figure 1 (grey area) which parts of the Netherlands were excluded. The target population excludes:

a. build-up areas, glasshouses, infrastructure (roads), water

b. soils with peat at about 20 to 25 cm below surface. So peat soils with a mineral topsoil (clay, sand) are NOT excluded.

c. naturally compact subsoils, such as boulder clay soils, 'knipkleigronden' and others

The area of the soils of category b, which are not overcompacted, is about 4.5%. The estimated total area (a, b and c) is about 5% to 6% of the area of the Netherlands. So correcton for this gives an estimate of about 43% instead of 45%. We corrected the percentage in the abstract and Conclusions, and added a sentence in the Results subsection 4.2 (first paragraph)

Finally, I was wondering at which depth soil samples were taken if no plough layer was present (if I understood correctly also uncultivated soils were sampled, right? page 4, line 28) and whether on cropland typically more compacted parts of the fields like traffic lanes or headland were sampled as well.

When it was not clear at what depth the subsoil was compacted, both in uncultivated and cultivated soils the decision was based on the penetration resistance as measured with a penetrometer. Sampling sites

were randomly selected, and selected sites in traffic lanes and headlands of arable fields were NOT excluded. We added a sentence in subsection 3.2, second paragraph.

Best regards,

Florian Schneider

[revised manuscript text omitted]